# Evaluation of Human Dental Pulp Stem Cells Expressing BMP2/7 Heterodimers in a Doxycycline-Inducible Manner

**DOI:** 10.3390/biom15121704

**Published:** 2025-12-06

**Authors:** Edit Hrubi, Ferenc Tóth, Gergely Nagy, József Tőzsér, Csaba Hegedűs

**Affiliations:** 1Department of Biomaterials and Prosthetic Dentistry, Faculty of Dentistry, University of Debrecen, H-4032 Debrecen, Hungary; 2Department of Biochemistry and Molecular Biology, Faculty of Medicine, University of Debrecen, H-4032 Debrecen, Hungary

**Keywords:** BMP2/7, DPSC, osteogenic differentiation, gene therapy, Tet-ON, bone regeneration

## Abstract

The BMP2/7 heterodimer is known as a stronger inducer of osteogenic differentiation than BMP2 or BMP7 homodimers. Our aim was to establish BMP2/7-expressing human dental pulp stem cells (DPSCs) to evaluate tReceived: 23 October 2025; Revised: 29 November 2025; Accepted: 3 December 2025; Published: 3 December 2025 he osteogenic potential of the genetically modified cells. Lentiviral transduction was used to introduce the Tet-ON-regulated transgene-containing vector to the cells. Endogenous heterodimers were detected at the mRNA and protein levels using RNA-seq, qPCR, and Western blot, while secreted heterodimers were detected using ELISA assays. Osteogenic differentiation was monitored by measuring alkaline phosphatase (ALP) activity, mineralization, and the expression levels of *RUNX2* and *ALPL* genes. Our results showed that ALP activity did not change in the transduced DPSCs; however, increased mineralization was detected, which correlates with the results obtained by RNA sequencing. Based on our results, BMP2/7-expressing DPSCs could be used in the treatment of bone defects, where heterodimers may have not only autocrine but also paracrine effects.

## 1. Introduction

Bone augmentation is a frequently used therapy in dentistry. The most commonly used surgical technique for this purpose is guided bone regeneration (GBR), in which barrier membranes and bone fillers are combined. Traditionally, allograft particles or autograft chips were used as a scaffold for mechanical support to avoid membrane collapse [1]. More recently, scaffold structures have been combined with mesenchymal stem cells for bone tissue reconstruction, an extensively investigated field of dental research. Stem cells show different degrees of osteogenic differentiation potential depending on the originating tissue. Among mesenchymal stem cell types, bone marrow stem cells (BMSCs) are the most commonly used and exhibit strong osteogenic differentiation capacity; however, their main limitation is the invasive nature of source tissue extraction. In contrast, human dental pulp tissue is readily available from extracted teeth. During osteogenic differentiation, dental pulp stem cells (DPSCs) express different osteogenic-related proteins, e.g., alkaline phosphatase, type I collagen, bone morphogenetic proteins (BMPs), osteonectin, osteopontin, and osteocalcin [2]. Among these, bone morphogenetic proteins belong to the TGFβ family and play key roles in both embryonic bone development and bone fracture healing [2]. BMPs have been shown to enhance osteogenic differentiation in vitro and in vivo. Furthermore, BMP2 and BMP7 homodimers are approved by the U.S. Food and Drug Administration (FDA) for clinical treatments aimed at bone regeneration. However, these treatments require high effective doses [3,4], and the controlled release of the growth factor from delivery systems has not yet been resolved [5]. At high doses, BMP homodimers may cause various side effects, such as activated bone resorption, ectopic bone formation, increased tumor formation risk, bone cysts, inflammatory and urogenital complications, and wound-related complications such as epidural hematoma, wound dehiscence, postoperative fever, and hemorrhage [4]. To date, no alternatives to BMPs have been identified that both effectively enhance bone healing and exhibit fewer side effects [6].

Besides BMP homodimers, the BMP2/7 heterodimer also plays an important role in the spatial regulation of the bone healing process [7]. The BMP2/7 heterodimer can induce in vitro osteogenic differentiation 15–20 times more efficiently than BMP2 or BMP7 homodimers alone, allowing for a reduction in the applied dose [7,8]. A low dose of the BMP heterodimer can induce a similar amount of bone formation as BMP homodimers but without inflammatory reactions [7,9]. Both homo- and heterodimer BMPs activate the same signaling pathways through binding to BMP receptor I (BMPRI) and BMP receptor II (BMPRII). BMP7 binds to BMPRI with low affinity and to BMPRII with high affinity, whereas BMP2 binds to BMPRI with high affinity and to BMPRII with low affinity [10]. It is hypothesized that the increased bioactivity of the BMP2/7 heterodimer results from its high-affinity binding to both BMPRI and BMPRII [10]. Another possibility is that the BMP2/7 heterodimer can upregulate BMP receptor genes [10]. A third hypothesis is that BMP homo- and heterodimers differentially regulate the synthesis of BMP antagonists and/or are differentially affected by these inhibitors [10]. It has been shown that the BMP2/7 heterodimer binds to the BMP antagonist Noggin—which prevents BMPs from binding to BMP receptors [10,11,12]—with lower affinity than BMP2 or 7 homodimers, resulting in a weaker inhibition of the heterodimer compared to the homodimer forms [12]. During the osteogenic differentiation of mesenchymal stem cells, Noggin levels decrease [13,14,15], suggesting that Noggin regulates BMP levels in differentiating cells [13,14,15,16].

When BMP2 and BMP7 proteins were transiently or constitutively co-expressed in A549 cells, BMP heterodimers were formed that induced bone-oriented differentiation and in vivo osteogenesis more effectively than corresponding homodimers [7,9,12,17]. However, in such a system, it is difficult to optimize the expression level of the individual BMP genes, and co-expression of the two BMPs also produces homodimers as by-products. In another study, a BMP2/7 fusion gene was created that encodes both BMP2 and BMP7 in parallel, connected by a linker. A549 producer cells were transduced with this fusion gene, and the resulting BMP2/7 heterodimer was applied to differentiate C2C12 myoblast cells. The heterodimer produced by the BMP2/7-modified cell line enhanced osteogenic differentiation more effectively than BMP2 or BMP7 homodimers alone and induced significantly lower levels of Noggin expression [12].

The aim of this study was to establish a DPSC-based bone tissue engineering system in which DPSC can efficiently express the BMP2/7 heterodimer in a regulated manner using a doxycycline-inducible Tet-ON expression vector. To identify the modified cells, green fluorescent protein (GFP) was co-expressed with the heterodimer from the same plasmid construct. Lentiviral transduction was used to introduce the transgene-containing vector into the cells, and heterodimer-expressing DPSCs were visualized by confocal laser scanning microscopy (CLSM) and measured by flow cytometry. Endogenous heterodimers were detected at the mRNA and protein levels by qPCR and Western blot, respectively, and secreted heterodimers were detected by ELISA. Osteogenic differentiation of BMP2/7-modified DPSCs was monitored by measuring alkaline phosphatase (ALP) activity, mineralization, and *RUNX2* and *ALPL* expression levels.

## 2. Materials and Methods

### 2.1. Cell Culture

DPSCs were isolated from the pulp tissue of healthy human wisdom teeth based on a published protocol [18]. The procedure was approved by the Regional and Institutional Ethics Committee of the University of Debrecen and authorized by the Hungarian National Public Health and Medical Officer Service (NPHMOS) (Approval code: IF-14135-8/2016; approval date: 03-NOV-2016). Informed consent was obtained from all individual participants included in the study (Patient declaration of agreement No. F0102/1ST). The donor was a 19-year-old male. DPSC cells were cultured to the fourth passage (P4) after isolation, then frozen in fetal bovine serum (FBS; Sigma-Aldrich, St. Louis, MO, USA, F9665) supplemented with 10% dimethyl sulfoxide (DMSO; Sigma-Aldrich, St. Louis, MO, USA, D8418) and kept in liquid nitrogen until use. P4 cells were used for lentiviral transduction, and the transduced cells were frozen at passage 8 (P8). P8 cells were used for all further experiments. Isolated DPSCs were cultured in Dulbecco’s modified Eagle’s medium/F12 (DMEM-F12; Thermo Fisher Scientific, Waltham, MA, USA, 11320033) supplemented with 10% fetal bovine serum (FBS; Sigma-Aldrich, St. Louis, MO, USA, F9665), 100 units/mL penicillin and 100 mg/mL streptomycin (Sigma-Aldrich, St. Louis, MO, USA, P0781), and 1% GlutaMAX (Life Technologies, Carlsbad, CA, USA, 10567014) at 37 °C and 5% CO_2_ in a humidified atmosphere. A 1:2 split ratio was used at each passage. In all figures, this medium is indicated as the control medium (CM). The osteo-inductive medium (OIM) was prepared by supplementing CM with 10 mM β-glycerophosphate (Sigma-Aldrich, St. Louis, MO, USA, G9891), 50 µg/mL of ascorbic acid (Sigma-Aldrich, St. Louis, MO, USA, 1043003), 0.1 µM of dexamethasone (Sigma-Aldrich, St. Louis, MO, USA, D4902), and 50 nM vitamin D3 (Sigma-Aldrich, St. Louis, MO, USA, 740292).

### 2.2. Construction of pTet-IRES-EGFP-BMP2/7 Plasmid

*BMP2* and *BMP7* gene sequences were amplified using Q5^®^ Hot Start High-Fidelity 2X Master Mix (New England BioLabs, Ipswich, MA, USA) with the following primers based on [12]:

For *BMP2*:

5′-attcgagctcggtacccgggatggtggccgggacccgctgtctt-3′;

5′-acttccacctccaccactaccacctcctccactacctccacctccacttcctccaccaccgcgacacccacaaccctccacaac-3′.

For *BMP7*:

5′-gtagtggtggaggtggaagtgacttcagcctggacaacgagg-3′;

5′-ataccgtcgagattctagagctagtggcagccacaggccc-3′.

The *BMP2* and *BMP7* coding fragments and the BamHI-digested pTet-IRES-EGFP backbone vector were purified after agarose gel electrophoresis with the Monarch^®^ DNA Gel Extraction Kit and assembled into pTet-IRES-EGFP-BMP2/7 using the NEBuilder^®^ HiFi DNA Assembly Cloning Kit (New England BioLabs, Ipswich, MA, USA) according to the manufacturer’s instructions (Appendix A).

### 2.3. Lentivirus Preparation and Transduction

Viral particles were produced in 293FT cells cultured in Dulbecco’s modified Eagle’s medium (DMEM) supplemented with 10% fetal bovine serum (FBS). Cells were grown to approximately 70% confluence, and 8 μg of pTet-IRES-EGFP-BMP2/7 and 8 μg of pLenti CMV rtTA3 Blast (w756-1) (Addgene plasmid 26429) were used to transiently transfect 293FT cells in a T75 flask using polyethyleneimine (Sigma). pLenti CMV rtTA3 Blast (w756-1), a gift from Eric Campeau, was co-transfected with 6 μg of psPAX2, 2 μg of pMD2.G, and 13 μg of salmon sperm DNA (Sigma Aldrich, St. Louis, MO, USA) in DMEM containing 1% FBS. The medium was replaced after 6 h. Conditioned media containing viral particles were collected after 3 days, clarified by centrifugation, and filtered through a 0.45 μm polyvinylidene fluoride (PVDF) filter (Millipore, Billerica, MA, USA), and stored at −70 °C.

For transduction, 500 μL of pLenti CMV rtTA3 Blast (w756-1) virus, 500 μL of pTet-IRES-EGFP-BMP2/7 virus, and 1 mL of fresh medium were mixed with 8 μg/mL of polybrene (Sigma Aldrich, St. Louis, MO, USA) in 6-well plates containing DPSC at 50% confluence. After overnight incubation, the medium was replaced. Medium containing 5 μg/mL of blasticidin (Thermo Fisher Scientific, Waltham, MA, USA) was added to the cells at passage one. After antibiotic selection, GFP and heterodimer expression was induced by the addition of 100 ng/mL of doxycycline (Appendix A), and the cells were examined under fluorescence microscopy (Zeiss Axiovert 100, Carl Zeiss Microscopy, Jena, Germany) and by flow cytometry (BD FACSCalibur, BD Biosciences, Franklin Lakes, NJ, USA) to determine transduction efficiency.

### 2.4. Confocal Laser Scanning Microscopy (CLSM)

Confocal images were taken using a FLUOVIEW FV 1000 confocal microscope (Olympus, Center Valley, PA, USA) based on an inverted IX-81 stand with a UPLS APO 60× (NA 1.35) oil immersion objective. GFP was excited with a 488 nm argon ion laser. PI was excited with a 543 nm HeNe laser.

### 2.5. Flow Cytometric Analysis

Flow cytometric measurements were performed using a Becton Dickinson FACSCalibur Instrument (San Jose, CA, USA). Data were collected and analyzed using CellQest Pro (version 5.1) software. GFP was excited with a 488 nm laser and fluorescence was detected in the FL1 channel using a 530/30 nm filter.

### 2.6. ELISA Assay

BMP2- and BMP7-specific ELISA assays were performed using the BMP-2 Quantikine ELISA kit (Bio-Techne R&D Systems Kft., Budapest, Hungary, DBP200) and the BMP-7 Quantikine ELISA kit (Bio-Techne R&D Systems Kft., Budapest, Hungary, DBP700), respectively, according to the manufacturer’s instructions.

### 2.7. BMP7 Western Blot

Overall, 10^5^ cells were seeded into Petri dishes and cultured for 6 days in CM in the presence of 0, 3.25, 6.25, 12.5, 25, 50, and 100 ng/µL of doxycycline. After 6 days, 10 µg/mL of Brefeldin A (Tocris Bioscience, Tocris, Bristol, UK) was added to the cells. Twenty-four hours later, the cells were lysed with radioimmunoprecipitation assay (RIPA) buffer, and the lysates were analyzed as described in ref. [19].

### 2.8. Osteogenic Differentiation

Osteogenic differentiation was performed as described in ref. [20]. A total of 30,000 cells were plated into each well of a 12-well culture plate (Sigma-Aldrich, St. Louis, MO, USA, Z707775) using 1 ml of CM. The cells were allowed to adhere overnight in CM, after which the medium was replaced with either CM or OIM, with or without 100 ng/ml doxycycline (DOX). The media were refreshed every second day

### 2.9. ALP Assay

ALP assay was performed as described in ref. [20]. Following 1, 2, or 3 weeks of osteogenic induction, cells were washed twice with 1 mL of 1× phosphate-buffered saline (PBS) and lysed in lysis buffer (10 mM Tris-HCl, pH 7.4; 100 mM NaCl; 1 mM ethylenediaminetetraacetic acid [EDTA]; 1% Triton X-100; 1% protease inhibitor cocktail [PIC]). Scraped lysates were transferred into sterile Eppendorf tubes. After centrifugation (10,000× *g*, 10 min, 4 °C), the supernatant was used to determine ALP activity and protein concentration (Pierce BCA Protein Assay, Thermo Scientific, Waltham, MA, USA, 23227). To determine ALP activity, 0.1% *p*-nitrophenyl phosphate (Sigma-Aldrich, St. Louis, MO, USA, N7653; in 0.1 M glycine, 1 mM MgCl_2_, ZnCl_2_, pH 10.4) was added to the samples, and absorbance was measured at 405 nm using a Hidex Sense Microplate reader. Kinetic measurements were performed every 3 min for 2 h.

### 2.10. Measurement of Calcium Deposition

The measurement of calcium deposition was performed as described in ref. [20]. After 3 weeks of osteogenic induction, cells were washed twice with 1 mL of 1× PBS (Sigma-Aldrich, St. Louis, MO, USA, P5493). Samples were fixed with 1 mL of ice-cold methanol (Sigma-Aldrich, St. Louis, MO, USA, 322415) for 30 min at room temperature. Afterward, samples were dried for 5 min at room temperature and then stained with 2% (*w*/*v*) Alizarin Red S (pH 7; Sigma-Aldrich, St. Louis, MO, USA, A5533). For quantification, Alizarin Red S–calcium complexes were extracted with 10% cetylpyridinium chloride (Sigma-Aldrich, St. Louis, MO, USA, C0732) diluted in 10 mM sodium phosphate buffer adjusted to a pH of 7 (Sigma-Aldrich, St. Louis, MO, USA, P5244), and absorbance was measured at 570 nm using a Hidex Sense Microplate reader. For standardization, the protein concentration of cell lysates was determined with the Pierce BCA Protein Assay (Thermo Scientific, Waltham, MA, USA, 23227) according to the manufacturer’s instructions from parallel samples that were not stained with Alizarin Red S. Calcium deposition is expressed as A570 nm/µg protein.

### 2.11. Total RNA Extraction, Reverse Transcription, and Real-Time Polymerase Chain Reaction (PCR)

Total RNA extraction, reverse transcription, and RT-PCR were performed as described in ref. [20]. Following 1, 2, or 3 weeks of osteogenic induction, cells were washed twice with 2 mL of 1× PBS. Total RNA was extracted with the Quick-RNA Miniprep Kit (Zymo Research, Irvine, CA, USA, R1054) according to the manufacturer’s instructions. The High-capacity cDNA Reverse Transcription Kit (Thermo Fisher Scientific, Waltham, MA, USA, 4368814) was used for reverse transcription using 0.5 µg of RNA per sample.

Gene expression levels were determined using TaqMan gene expression assays for *RUNX2* (Applied Biosystems, Waltham, MA, USA, Hs00231692_m1), *BMP2* (Applied Biosystems, Waltham, MA, USA, Hs00154192_m1), *BMP7* (Applied Biosystems, Waltham, MA, USA, Hs), *ALPL* (Applied Biosystems, Waltham, MA, USA, Hs01029144_m1), and *NOG* (Applied Biosystems, Waltham, MA, USA, Hs00608272_m1) with 5x HOT FIREPOL Probe qPCR Mix Plus (no ROX; Solis BioDyne, Tartu, Estonia, 08-15-00001). Expression levels were normalized to the reference housekeeping gene glyceraldehyde-3-phosphate dehydrogenase (*GAPDH*; Applied Biosystems, Waltham, MA, USA, Hs02758991).

### 2.12. RNA-Seq

To generate whole-transcriptome data, high-throughput next-generation mRNA sequencing was performed on an Illumina platform. The RNA sequencing library was prepared using whole RNA isolates with the Ultra II RNA Sample Prep kit (New England BioLabs, Ipswich, MA, USA) according to the manufacturer’s instructions.

Next-generation sequencing was performed on an Illumina NextSeq 500 instrument with a single-end 75-cycle run following the quality control of the RNA sequencing libraries.

### 2.13. RNA-Seq Analysis

Raw sequence reads were aligned to the hg19 human reference genome assembly using hisat2 v2.1.0 (24). BAM files were created with SAMtools v1.7 [20]. Gene expression levels, determined in fragments per kilobase per million mapped fragments (FPKMs), were calculated using StringTie v1.3.4d [21], and the resulting FPKM values were decile-normalized. Protein-coding genes were selected using the Ensembl database (Ensembl BioMart-release 75). Dot and line plots were visualized with GraphPad Prism v8.0.1.

Gene ontology (GO) analyses were performed using ShinyGO v0.85 [22].

### 2.14. Statistical Analysis

Error bars in the figures represent the standard deviation calculated from three parallel measurements (technical replicates). ANOVA followed by the Bonferroni post hoc test was used for multiple comparisons.

## 3. Results

An increasing number of GFP-positive cells was observed when transduced DPSCs were visualized by confocal microscopy (Figure 1A) and analyzed by flow cytometry (Figure 1B–D) after 7 days of treatment with increasing concentrations of doxycycline. GFP fluorescence intensity increased proportionally with the doxycycline concentration (Figure 1C), whereas the percentage of GFP-positive cells reached a plateau at 25 ng/mL doxycycline (Figure 1D).

Endogenous heterodimer production was analyzed by Western blot using a BMP7-specific antibody on whole-cell lysates from transduced DPSCs treated for 3 days with increasing concentrations of doxycycline. A doxycycline concentration-dependent increase in heterodimer levels was detected in the lysates (Figure 2A,B), consistent with the CLSM and FACS analyses. Based on these results, 100 ng/mL of doxycycline was used in further experiments, as this concentration yielded the highest GFP fluorescence intensity—proportional to the BMP2/7 expression—the highest viral transduction efficiency (50%) measured by FACS, and the highest endogenous heterodimer levels detected by Western blot.

Secreted heterodimer concentrations were determined from the culture media of transduced DPSCs treated with 100 ng/mL of doxycycline for one, two, or three days using BMP2- or BMP7-specific ELISA assays. The assay was performed on DPSCs treated with doxycycline in CM or OIM cell culture media. In both media, the concentration of secreted heterodimer significantly increased after three days (Figure 2C,D). CLSM, FACS, Western blot, and ELISA analyses indicate that the doxycycline-inducible BMP2/7 heterodimer expression system functions as expected.

Bone-oriented differentiation was assessed by measuring alkaline phosphatase enzyme activity in lysates from DPSCs treated with 100 ng/mL of doxycycline for one, two, or three weeks in CM or OIM. Significantly increased ALP activity was detected in OIM and OIM supplemented with doxycycline (OIM + DOX) compared with CM used as a negative control; however, no difference was observed between OIM and OIM + DOX. Interestingly, ALP activity decreased in DPSCs cultured in CM supplemented with doxycycline after two and three weeks (Figure 2E).

Osteogenic differentiation of transduced DPSCs was further characterized by measuring mineral deposition in the extracellular matrix. Alizarin Red S-stained hydroxyapatite crystals were measured spectrophotometrically after one, two, or three weeks of doxycycline treatment in CM or in OIM. Significantly increased mineralization was measured in DPSCs cultured in OIM + DOX compared to OIM, CM, or CM + DOX (Figure 2F,G).

The expression of genes involved in osteogenic differentiation was also measured in transduced DPSCs after one, two, or three weeks of doxycycline treatment. Among these, *RUNX2* and *ALPL* were selected as markers expressing at different stages of osteogenic differentiation: high *RUNX2* expression is characteristic of preosteoblasts, whereas *ALPL* expression peaks in mature osteoblasts. *RUNX2* levels increased after one week, and *ALPL* levels increased after two weeks in OIM and OIM + DOX compared with CM, as expected (Figure 3A, B). No differences were detected between CM and CM + DOX in the case of *RUNX2* (Figure 3A) or between OIM and OIM + DOX for either *RUNX2* or *ALPL* (Figure 3A,B). Notably, *ALPL* expression was significantly higher in CM + DOX than in CM (Figure 3B). The decreased levels of *RUNX2* observed after two or three weeks in CM + DOX, OIM, and OIM + DOX compared with CM (Figure 3A) also indicate the progression of osteogenic differentiation, which was further reflected in the reduced expression of the BMP antagonist Noggin (*NOG*) (Figure 3C). As a control of the expression system, *BMP2* (Figure 3D) and *BMP7* (Figure 3E) expression levels were also measured and were significantly increased in CM + DOX and OIM + DOX compared to CM or OIM, respectively, after one, two, or three weeks, as expected. Interestingly, a significant decrease in *BMP2* and *BMP7* was observed in OIM + DOX after two or three weeks compared with one week of doxycycline treatment (Figure 3D,E), something which was not seen in CM + DOX.

Comparison of RNA expression profiles between the treated (OIM + DOX) and untreated (OIM) DPSC-BMP2/7 cell cultures after 3 weeks revealed a total of 280 differentially expressed genes (fold change > 1.5; Figure 3F and Appendix A).

These genes were further compared to a specific list containing only genes encoding proteins involved in BMP signaling or pathways connected to its regulation. Of the 25 genes identified in this analysis (Figure 3G), only 3 (*ID1, BAMBI,* and *FGF13*) were upregulated, whereas 27 *(FBN1*, *FBN2*, *ADAMTS2, ADAMTS7, ADAMTS12*, *HSPG2*, *NEDD4, ALPL, DDR2, ZFYVE16, SATB2, NEO1, MAPK7, NRP2, PIK3CD, HDAC6, SMURF1, LRP6, MTOR, APC, JAG1, EP300,* and *CREBBP*) were downregulated in treated cells compared with untreated cells.

Another analysis of the differentially expressed *BMP*s and their receptors revealed that, besides the overexpression of BMP2/7, two other *BMP*s involved in osteoblast differentiation or mineralization (*BMP6* and *BMP8A*) were also upregulated in the treated cells. In contrast, the expression of the corresponding *BMP* receptors was downregulated compared to the untreated cells; however, the fold changes were not remarkable (Figure 3H).

Gene Ontology (GO) enrichment analysis of molecular function terms revealed several significantly overrepresented categories. The highest fold enrichment was observed for proteoglycan binding and extracellular matrix structural constituent functions, followed by growth factor binding. Additional enriched terms included integrin binding, actin filamentbinding, heparin binding, glycosaminoglycan binding, and cadherin binding. Several adhesion-related categories, such as cell adhesion molecule binding and structural molecule activity, were also significantly represented. Lower but still notable enrichments were detected for actin binding, calcium ion binding, protein-containing complex binding, cytoskeletal protein binding, andsignaling receptor binding. The number of genes contributing to each category ranged from approximately 10 to over 40, and the statistical significance of enrichment (−log10 FDR) varied across terms, with the strongest signals observed for proteoglycan- and ECM-related functions.

## 4. Discussion

The BMP2/7 heterodimer is described as a stronger inducer of osteogenic differentiation than the BMP2 or BMP7 homodimers [7,8,23,24,25]; however, its effect on DPSCs has not yet been investigated. In this study, we evaluated the osteogenic differentiation potential of DPSCs expressing the BMP2/7 heterodimer in a controlled manner using a lentiviral expression system. Inducible lentiviral systems offer powerful tools for gene expression studies. In recombination-based approaches (Cre-Lox and FLP-FRT systems), the activation or knockout of the target gene is irreversible, whereas in tetracycline (Tet) and estrogen receptor-inducible expression systems, it is reversible. The Tet system has very tight control of expression and pleiotropy is minimized because the Tet-ON sequence is missing from mammalian cells, and the level of gene expression can be adjusted by varying the concentration of tetracycline or doxycycline [26].

In our system, a Tet-ON-controlled expression vector was introduced into human DPSCs using a lentiviral transduction system, enabling dose-dependent BMP2/7 expression in response to doxycycline. BMP2/7 was co-expressed together with GFP from the same plasmid construct, allowing for the successful detection of transduced cells by fluorescence imaging or flow cytometry. BMP2/7 expression in genetically modified DPSCs increased in a concentration-dependent manner and was measurable at the mRNA level by qPCR using *BMP2*- or *BMP7*-specific TaqMan assays and by RNA sequencing. At the protein level, BMP2/7 expression was detected by Western blot and ELISA. For the Western blots, BMP2/7 appeared as a sharp band at an expected heterodimer size using a BMP7-specific antibody, with the intensity increasing proportionally with the doxycycline concentration. Secreted BMP2/7 was also quantified by ELISA. Notably, the concentration measured by the secreted BMP2/7 was 15–20 times lower than that measured by the BMP2-specific versus the BMP7-specific ELISA, suggesting that epitopes recognized by the BMP2-specific antibody may be partially masked in the heterodimer, thereby reducing the binding affinity in both ELISA and Western blot analyses.

DPSCs are known to undergo spontaneous differentiation during long-term cultivation in medium lacking osteoinductive factors (CM). This may occur due to culture conditions or heterogeneity within the isolated cell population in terms of the osteogenic potential [27,28,29]. This spontaneous differentiation was reflected by the steady increase in *ALPL* gene expression after 1, 2, and 3 weeks. When the heterodimer was expressed in cells cultured in CM + DOX, the secreted BMP2/7 interacted with BMP receptors and upregulated osteogenic genes such as *ALPL*, explaining the increased *ALPL* levels after 1 and 2 weeks in CM + DOX compared with CM. In our previous study [30], we demonstrated that BMP2 together with a phosphate source (β-glycerophosphate) is sufficient to induce osteogenic differentiation and mineralization without additional osteogenic factors such as those present in OIM medium (dexamethasone, ascorbic acid, and vitamin D3). These findings suggest that BMP2/7 can also induce osteogenic differentiation on its own, which is consistent with our current results. Culture conditions can also influence the spontaneous differentiation of DPSCs; however, to date, no optimal culture medium that prevents spontaneous differentiation while preserving stem cell properties has been reported [28].

With regard to ALP activity, no difference could be measured between DOX-induced and non-induced cells in OIM, while a slight but significant increase in mineralization was observed, reflecting a positive effect of BMP2/7 in the late stage of osteogenic differentiation. Interestingly, under CM + DOX conditions, ALP activity decreased at weeks 2 and 3 compared with CM, despite the concomitant upregulation of *ALPL* mRNA. This apparent discrepancy may be due to the absence of osteoinductive supplements (OIM). BMP2/7 induction alone may prime the cells toward an osteogenic lineage without promoting full maturation, resulting in increased *ALPL* gene expression but insufficient enzymatic activity. This observation is consistent with our previous study on DPSCs treated with BMP2 [30]. Because CM medium lacks a phosphate source necessary for mineralization, calcium deposition does not occur in cells cultured in CM or CM + DOX, as expected.

When heterodimer expression was validated by qPCR, lower *BMP2* and *BMP7* expression levels were observed in OIM + DOX at weeks 2 and 3 compared with week 1, rather than the continuous increase seen in CM + DOX. This may result from self-regulation through a negative feedback loop, as reported in previous studies [31,32]. Cells cultured in OIM + DOX undergo a complete differentiation than those cultured in CM + DOX, as also reflected by the decreased expression of the BMP antagonist *NOG* in OIM + DOX relative to CM + DOX, suggesting that the negative feedback mechanism is activated under these conditions.

*NOG* expression in cells cultured in OIM was significantly lower compared to CM, reflecting osteogenic differentiation, as NOG is a known inhibitor of this process through its role as a BMP antagonist [33,34]. When heterodimer expression was induced in CM + DOX-treated cells, *NOG* expression decreased, suggesting that the heterodimer alone can induce osteogenic differentiation, which is consistent with our previous study demonstrating that BMP2 alone is sufficient to trigger osteogenic differentiation [30]. A more pronounced decrease in *NOG* expression was observed in cells cultured in OIM or OIM + DOX, indicating more effective induction of osteogenic differentiation, as expected.

Among the products of osteogenic differentiation-related genes showing increased expression levels in BMP2/7-expressing cells, ID1 has previously been described as a regulator of DPSC osteogenic differentiation by affecting the availability of dimerization partners for TWIST1. Overexpression of ID1 in DPSCs increased mineralization in the absence of elevated ALP activity [35], in line with our observations. BMP8A, which is induced at the onset of mineralization in KS483 preosteoblast cells [36], was also increased in OIM + DOX-treated DPSCs. Several negatively regulated genes that exhibited a 2-fold decrease in expression in OIM + DOX compared with OIM also support the positive effect of BMP2/7. Among these, fibronectin 1 (FN1) is involved in the regulation of osteoblast differentiation through the activation of the WNT/β-catenin signaling pathway [37,38]. Yang et al. showed that osteoblastic marker gene expression and extracellular matrix mineralization were significantly enhanced following FN1 overexpression and decreased after FN1 knockdown in MC3T3-E1 cells; however, the negative effect of FN1 knockdown was reversed by ITGB1 downregulation [39], which was also decreased in our experiments to a similar extent as FN1. Fibrillins (FBN1 and FBN2) are extracellular matrix compounds responsible for the bioavailability of TGFβ and BMPs. In *Fbn1*-/- mice, osteoblasts mature more rapidly due to increased availability of matrix-bound BMPs [40,41], in line with our observation that BMP2/7-expressing DPSCs show increased mineralization potential. Heparan sulfate (HSPG) is a cell surface proteoglycan that directly regulates BMP signaling by sequestering BMP2 and mediating its internalization; depletion of HSPG enhances BMP2 bioactivity during osteogenic differentiation of C2C12 cells [42]. Reduced expression of type 1 BMP receptors BMPR1A and ACVR1, which can bind BMP2/7 heterodimers with higher affinity than homodimers [43] and are expressed in the DPSCs used in this study, could also be a sign of the progressed state of differentiation, as type 1 receptor expression has been shown to decrease during osteogenic differentiation of BMSCs [44]. Possible mechanisms underlying the positive effect of BMP2/7 on the osteogenic differentiation of transduced DPSCs are summarized on Figure 4.

A deeper analysis of RNA sequencing data revealed that, among the 280 differentially expressed genes, a subset (*BAMBI*, *FBN1*, *FBN2 HSPG*, *NEDD4*, *HDAC6*, *SMURF1*, *SMAD6*, *MTOR*, and *APC*) negatively regulates osteogenic differentiation, whereas another subset (*ID2*, *CREBBP*, *EP300*, *JAG1*, *LRP6*, *NRP2*, *MAPK7*, *NEO1*, *SATB2*, *ZFYVE16*, *ALPL*, *DDR2*, *FN1*, and *PIK3CD*) accelerates the process. Only *ID1*, *BAMBI*, *BMP6*, and *BMP8A* were upregulated. Because RNA sequencing results represent a single time point within a 3-week differentiation process, these data do not allow us to determine whether the overall effect of BMP2/7 on the osteogenic differentiation of transduced DPSCs was positive or negative.

Taken together, GO molecular function analysis of differentially expressed genes indicated a strong enrichment of categories associated with extracellular matrix organization, cell adhesion, growth factor interaction, and cytoskeletal regulation. These functions are characteristic of cells undergoing osteogenic commitment and matrix-producing activity. Although the enrichment data provide indirect evidence, the overall pattern is highly consistent with a shift toward osteogenic differentiation. Thus, the results support the interpretation that the analyzed gene set reflects activation of molecular pathways linked to osteoblast development and mineralized matrix formation.

## 5. Conclusions

Although DPSCs overexpressing BMP2/7 showed a minor enhancement in matrix maturation, the extent of this change was smaller than expected based on the previously reported strong osteogenic differentiation activity of BMP2/7 [8,23,24]. This discrepancy may be attributable to the heterogeneity among DPSCs isolated from different donors [27]. The osteogenic differentiation ability of DPSCs strongly depends on their metabolic signature and the expression of TGF-β pathway and cytoskeletal protein genes, which define rapid-aging and slow-aging DPSC phenotypes; the rapid-aging phenotype exhibits reduced differentiation potential [27]. This raises the possibility that BMP2/7 overexpression may have different effects on rapid- versus slow-aging DPSCs, which warrants further analysis. Negative feedback mechanisms may also be responsible for the modest effect of BMP2/7, and this could potentially be mitigated by decreasing doxycycline concentrations to fine-tune heterodimer expression. A limitation of this study is that cells from only a single donor were used, providing just one biological replicate. While this allowed for detailed mechanistic investigation under controlled conditions, it restricted the generalizability of the findings. Previous studies have highlighted variability between human DPSC clones isolated from the same donor, as well as between DPSC populations isolated from different donors, affecting the proliferation, differentiation potential, and response to osteogenic induction [27,45]. Therefore, the validation of these results in additional donors will be essential to confirm the reproducibility and broader applicability of the observed effects. BMP2/7-expressing DPSCs should also be tested under in vivo conditions, where the heterodimer produced by transduced cells could exert not only autocrine effects on DPSCs but also paracrine effects on other stem or preosteoblast cells recruited to the bone defect area.

## Figures and Tables

**Figure 1 biomolecules-15-01704-f001:**
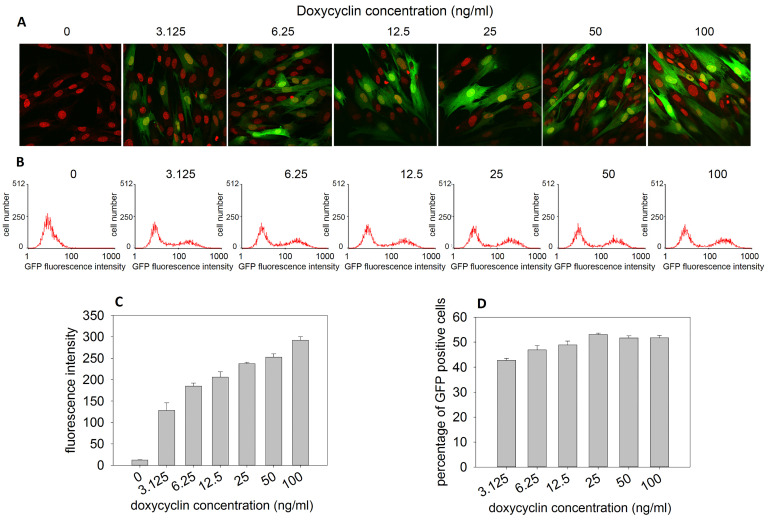
CLSM and flow cytometric analyses of DPSCs transduced with BMP2/7 heterodimer and GFP Tet-ON system. (**A**) Representative CLSM images of transduced DPSCs after 7 days of treatment with increasing concentrations of doxycycline. (Green indicates GFP, while red indicates nuclei stained with propidium iodide.) (**B**) Flow cytometric analysis of transduced DPSCs after 7 days of treatment with increasing concentrations of doxycycline. Histograms show the distribution of GFP fluorescence intensity at each doxycycline concentration. (**C**) Mean GFP fluorescence intensities gated on GFP-positive cells at different doxycycline concentrations, measured by flow cytometry. (**D**) Percentage of GFP-positive cells at different doxycycline concentrations, measured by flow cytometry. The height of the bars represents the mean, and the error bars represent the standard deviation of three parallel measurements.

**Figure 2 biomolecules-15-01704-f002:**
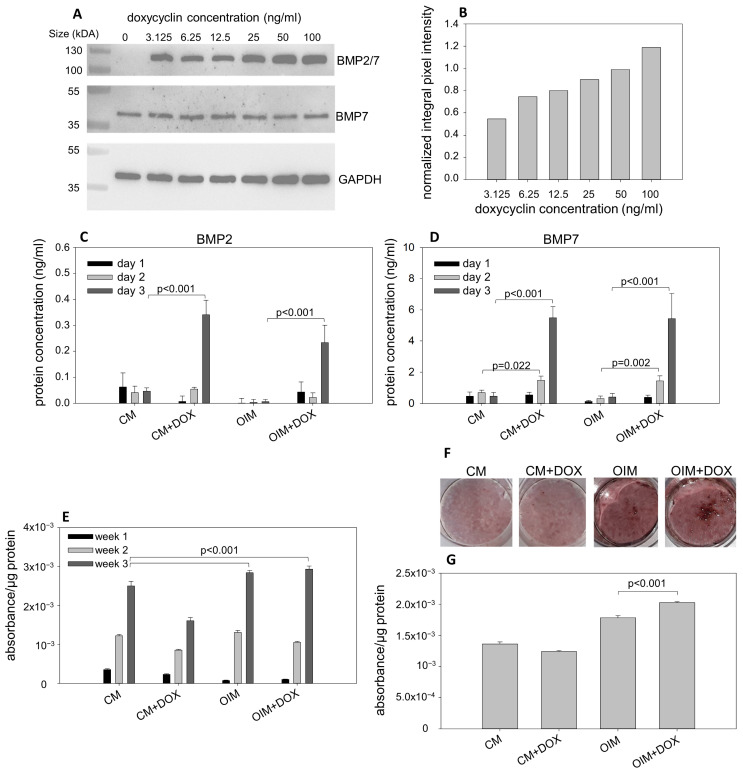
Western blot analysis, ELISA analysis, examination of ALP activity, and mineralization of transduced DPSCs. (**A**) Whole-cell lysates from transduced DPSCs treated for 3 days with increasing concentrations of doxycycline were analyzed by Western blot. The BMP2/7 heterodimer and the endogenous BMP7 were detected using a BMP7-specific antibody (Appendix A shows the whole gels). (**B**) Quantification of Western blot bands. Integral pixel intensities normalized to GAPDH are shown in the bar chart. (**C**) Quantitative analysis of secreted BMP2/7 heterodimer using BMP2-specific ELISA in culture media from cells grown in CM, CM + DOX, OIM, or OIM + DOX for one, two, or three days. (**D**) Quantitative analysis of secreted BMP2/7 heterodimer using BMP7-specific ELISA in culture media from cells grown in CM, CM + DOX, OIM, or OIM + DOX for one, two, or three days. (**E**) ALP activity of transduced DPSCs cultured in CM, CM + DOX, OIM, and OIM + DOX for one, two, or three weeks. (**F**) Visualization of calcium deposits of transduced DPSCs cultured in CM, CM + DOX, OIM, and OIM + DOX for three weeks. (**G**) Quantitative analysis of calcium-bound dye based measured by spectrophotometry. The height of the bars represents the mean, and the error bars represent the standard deviation of three parallel measurements. One-way ANOVA, combined with the Bonferroni post hoc test, was used for statistical analysis.

**Figure 3 biomolecules-15-01704-f003:**
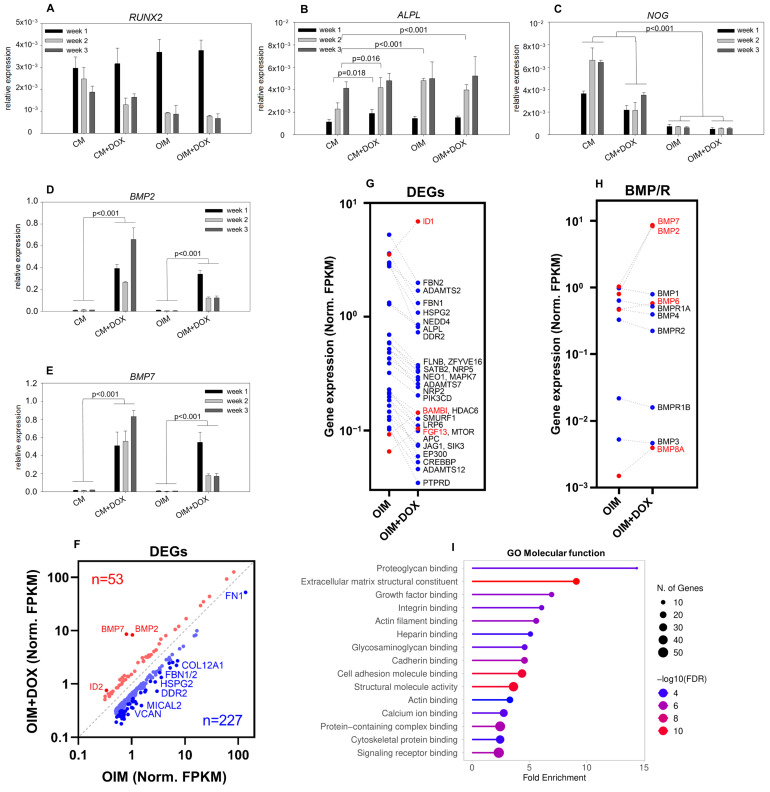
RT-qPCRanalysis of transduced DPSCs and differentially expressed genes in BMP2/7 heterodimer-expressing cells. (**A**) *RUNX2* expression in transduced DPSCs cultured in CM, CM + DOX, OIM, and OIM + DOX for one, two, or three weeks. (**B**) *ALPL* expression in transduced DPSCs cultured in CM, CM + DOX, OIM, and OIM + DOX for one, two, or three weeks. (**C**) *NOG* expression in transduced DPSCs cultured in CM, CM + DOX, OIM, and OIM + DOX for one, two, or three weeks. (**D**) *BMP2* expression in transduced DPSCs cultured in CM, CM + DOX, OIM, and OIM + DOX for one, two, or three weeks. (**E**) *BMP7* expression in transduced DPSCs cultured in CM, CM + DOX, OIM, and OIM + DOX for one, two, or three weeks. (**F**) Differentially expressed genes (DEGs) that up- or downregulated upon BMP2/7 overexpression (+DOX). Red dots show upregulated genes while blue dots show downregulated genes (fold difference (FD) > 2, red; 2 > FD > 1.5, light red; FD < 1/2, blue; 1/2 < FD < 2/3, light blue). (**G**,**H**) Changes in the expression of osteogenic differentiation-related genes, and *BMP* and *BMPR* genes in BMP2/7-expressing DPSCs. Red dots show upregulated genes while blue dots show downregulated genes. Dot-and-line plots represent normalized FPKM values of osteogenesis-related differentially expressed genes (DEGs) (**G**) and *BMP/BMPR* genes (**H**) upon BMP2/7 overexpression (+DOX). (**I**) GO terms (molecular functions) enriched in the DEGs. The height of the bars represents the mean, and the error bars represent the standard deviation of three parallel measurements. One-way ANOVA with the Bonferroni post hoc test was used for statistical analysis.

**Figure 4 biomolecules-15-01704-f004:**
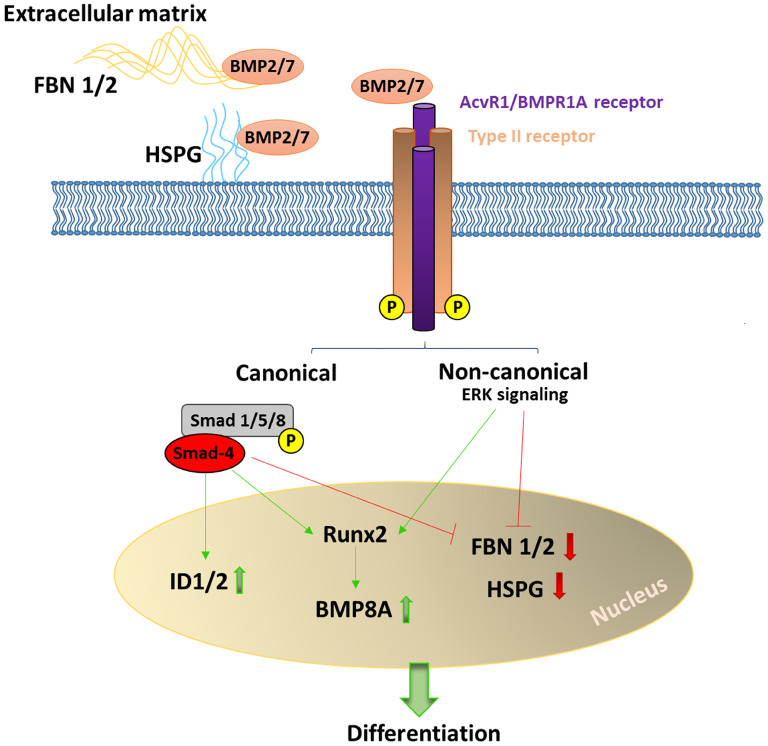
Possible mechanisms of BMP2/7 heterodimer action on transduced DPSCs. Among the genes differentially expressed in response to BMP2/7 heterodimer action in DPSCs, the increase in *ID1*, *ID2*, and *BMP8A* and the decrease in *FBN1/2* and *HSPG* represent changes that support enhanced differentiation (Green arrows indicate gene activation and red arrows indicate gene repression).

## Data Availability

The RNA-seq datasets generated during the current study are openly available in the NCBI SRA repository, under the BioProject accession number: PRJNA1212899. All other datasets generated during and/or analyzed the current study are available from the corresponding author upon reasonable request.

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
