# Peer review of "Evaluation of Human Dental Pulp Stem Cells Expressing BMP2/7 Heterodimers in a Doxycycline-Inducible Manner"

_biomolecules, 2025, doi:10.3390/biom15121704_

Round 1
Reviewer 1 Report
Comments and Suggestions for Authors
The authors have a good system to test if the expression of the trangene for BMP2/7 heterodimer leads to more efficient osteogenic differentiation from human dental pulp stem cells. This has been verified in other cell lines but has not specifically been done in the human DPSCs.
- Lines 309-310- Discuss the steady increase in ALPL in both the CM and CM+DOX group based on previously published studies. What is driving this increase in the absence of induction?
- Lines 317-318- Significant decrease in BMP2 and BMP7 is observed in the OIM+DOX condition. This is not what is expected. Discuss this observation and explain why this might be the case based on previous studies. Is this due to the increased mineralization?
- Perform a Gene Ontology analysis of the top 280 differentially expressed genes and include this in Figure 3. This will allow readers to visualize the pathways these genes contribute to.
- Discussion- Highlight the limitation of the study-one donor was use, hence just one biological replicate.
- Authors switch between using 'DM' and 'OIM'. Please stick to one acronym throughout.
Reviewer 2 Report
Comments and Suggestions for Authors
The manuscript titled ”Evaluation of human dental pulp stem cells expressing BMP2/7 heterodimer in a doxycycline inducible manner” delves into the effect for expressing BMP2/7 heterodimer to human dental pulp stem cells on osteogenic potential. Through their research, the authors noted that the assessment of osteogenic differentiation with ALP activity, calcium deposition and osteogenic related gene expression. This is an elaborate study design, however there are some questions that need to be answered to improve the manuscript.
- In terms of CM+DOX group on Figure2, it has increase on BMP2 and BMP7 on day 3 with ELISA, however the result of ALP activity is decrease compared to other group, especially week 3. Also, the result of measuring calcium is not increase. The authors should describe this point on the discussion part.
- In Figure 3C, NOG expression on DM group (probably OIM group) is significantly decrease compared to CM group, however both groups are low on BMP2 and BMP7 expression in figure 3D and 3E. Why is the difference? The authors should describe.
- Figure 2C,D,E,F,G: Shown groups on each figure are not matched with figure legend, especially DM and DM+DOX groups. Authors should check them.
- Figure 3A,B,C,D,E: Shown groups on each figure are not also matched with figure legend, especially DM and DM+DOX groups. Authors should check them.
- Line 176: Reference number should be described after “Toth et al.,2021.”
- Line 194: “2.10.Measurement of calcium deposition” section doesn’t describe the period for osteogenic induction. Probably calcification need a limited period of time. Please add it.
- Line 206: “2.11.Total RNA extraction, reverse transcription, and real-time polymerase chain reaction” section lacks the description of NOG and BMP7. Please add them.
- Figure 3G,H: Size are too small. It is hard to look at each gene expression. Please change the large size.
Round 2
Reviewer 2 Report
Comments and Suggestions for Authors
The manuscript is now much improved.